# Care Dependency of Hospitalized Stroke Patients Based on Family Caregivers’ and Nurses’ Assessments: A Comparative Study

**DOI:** 10.3390/healthcare10061007

**Published:** 2022-05-30

**Authors:** Nursiswati Nursiswati, Ruud J. G. Halfens, Christa Lohrmann

**Affiliations:** 1Institute of Nursing Science, Medical University of Graz, 8010 Graz, Austria; christa.lohrmann@medunigraz.at; 2Department of Health Services Research, Maastricht University, 6229 GT Maastricht, The Netherlands; r.halfens@maastrichtuniversity.nl

**Keywords:** care, caregivers, dependence, nurse, stroke

## Abstract

Stroke impacts care dependency, and thus the patient needs home care after suffering a stroke. This study was carried out to investigate similarities and differences between the assessments made by family caregivers and nurses regarding the care dependency level of stroke patients in Indonesian hospitals. This study was a comparative study of the care dependency of stroke patients. Data were collected on the stroke wards on the day of admission using the Care Dependency Scale (CDS). The sample consisted of 118 family caregivers and 21 nurses. The Wilcoxon signed-rank test was performed to determine the mean differences between the paired data collected by family caregivers and nurses. The results of this study show that significant differences exist between the family caregivers’ and nurses’ assessments regarding the care dependency levels of stroke patients. Nurses assigned higher scores to all CDS items than family caregivers did. Significant differences between the family caregivers’ and nurses’ assessments were observed on numerous items of the Care Dependency Scale. This study contributes to efforts to raise awareness of potential differences in perceived care dependency levels of stroke patients. The findings can help nurses plan the patient’s discharge together with family caregivers.

## 1. Introduction

Stroke is a global burden. Every year stroke affects 15 million people worldwide, with approximately 5.9 million cases resulting in death [1]. The burden of stroke in terms of mortality, morbidity and disability is increasing across the world. The prevalence of people suffering from a stroke has steadily increased in Indonesia from 7% to 10.9% between 2013 and 2018 [2]. About 44% of stroke survivors depend on their family for daily activities, and the dependency level has not been shown to decrease within six months after discharge from a stroke unit.

Family support for a stroke patient is indispensable, helping patients cope with physical and psychosocial impairments. This support can begin as early as during the acute care phase in the hospital setting and continue after stroke patients return home. The question of whether family caregivers should be allowed to take part in (stroke) caregiving during hospitalization is has cultural aspects. For instance, it is common in Indonesia for family caregivers to be obligated to care for their family members [3]. In Indonesia, family caregivers are allowed to stay beside the patient´s bed during the day or night and participate in stroke care. Previous research has revealed that the participation of the family caregivers in stroke care helps them understand the physical and psychosocial changes effected by the stroke [4]. Gaining knowledge about the bio-psycho-social changes related to stroke can decrease family caregivers’ anxiety [5], reduce their uncertainty about the future [6], help give them a more positive view of their caregiving experience [7] and support improved discharge planning [8]. 

Despite the existence of benefits from family participation in the post-stroke care process, differences of family caregivers’ assumptions regarding the patient’s care needs compared to nurses’ assessment can happen. It is no doubt that stroke consequences lead to family caregiver burden, and the burden is influenced by objective and subjective components. The objective components are the demands caused by the degree of care dependency of their care recipients. Meanwhile the subjective components include the way a caregiver perceives the caregiving task [9]. Additionally, the burden of family caregiver was due to insufficient social support [10]. 

A different situation, having a professional background with the norms and regulations of standardized care, better knowledge about stroke, and absence of psychological burden might play a role in the nurses’ assessment. With regard to assessing a stroke patient’s care dependency, health professionals including nurses are experts in terms of norms or a desirable standard. Nurses’ education, experiences, abilities, prior learning, attitudes and values play a critical role in how such assessments are made [11]. Nevertheless, continuing education, for example, training in a work environment is more valuable [12], because this can increase knowledge and abilities to sustain the quality of care. These different situations and backgrounds can result in a mismatch regarding the degree of support and help from those two caregiver groups and prevent adequate discharge planning for stroke patients. The family caregivers’ perception of the health and functional status of their care recipients is one of the important domains for partnering with nurses and other health professionals. Studies comparing nurse with family caregiver assessments care dependency are scarce, specifically for assessing stroke patients. Therefore, a comparative study, a finding can be seen whether there are differences or similarities in the assessment between the two caregiver groups. 

However, all provided assistance and care should be tailored to meet stroke patients’ needs and be based on the degree of care dependency. Care dependency defined as “a process in which the professional offers support to the patient whose self-care abilities have decreased and whose care demands make him/her to a certain degree dependent” [13]. The Care Dependency Scale (CDS) is a valid and reliable tool that has been developed for assessing nursing care needs in a comprehensive way [13]. 

Nurses and other professional care providers are responsible for assessing a patient’s care dependency level in the hospital in order to offer appropriate help and support the patient’s independence. By receiving additional input during the assessment process from patients and the proxy assessment, nurses can promote professional, shared decision-making [14] and support a sustainable health-promoting process [15]. This is an especially vital component of successful post-stroke care in Indonesia, as this care is mostly provided at home by responsible family caregivers. The situation in Indonesia differs from that in Western countries where the majority of hospitals have special post-stroke rehabilitation care wards. When stroke patients do not have access to rehabilitation care, the support of family caregivers is essential. It has been assumed that family caregivers can use the CDS as part of their caregiving process at home, especially when the degree of care dependency remains unchanged or does not improve significantly within one month of the patient’s discharge from the hospital [16]. 

A previous study revealed that the care dependency assessments of primary nurses and team nurses differed [13]. Another related research project on rater perspectives showed that nurse practitioners and care recipients differed in terms of their perceptions of the sufficiency of care offered to meet stroke rehabilitation needs [17]. These findings indicate that a comparison between family caregivers’ and nurses’ assessments of the care dependency of stroke patients could be meaningful for many reasons. 

First, few studies have been carried out on family caregivers’ assessments of patients’ levels of care dependency. Nurses can find out how a family caregiver defines care dependency and care needs by determining whether this assessment is similar or different from theirs. Second, the family caregivers’ assessments can improve the quality of patient-tailored stroke care, because neither too little or excessive assistance is offered. By determining the care dependency score of the patient, a family caregiver can provide assistance based on the concrete needs of their loved ones. Therefore, it is important to get agreement between the nurses’ and family caregivers’ assessments of the care dependency levels in the future when the CDS is applied in daily practice. According to Borgata, dependency can be influenced by both an individual and the person’s environment [18], for example, a stroke patient’s nurse or family caregiver and unsafe home interiors that resulted in functional limitation. If caregivers offer too much assistance, they might undermine the patient´s independence, but if they offer too little assistance, the patient’s needs are not met. For this reason, an understanding of the similarities and differences between care dependency scores assigned during nurses’ and family caregivers’ assessments is warranted. 

## 2. Materials and Methods

### 2.1. Study Design

The present study is a descriptive, comparative study of the care dependency levels of stroke patients in the Indonesian hospital setting.

### 2.2. Participants

This study was conducted on five stroke wards at four participating hospitals on the island of Java, Indonesia. We calculated that 125 stroke patients were needed by using the formula (*n* = t2pq/d2). We calculated the effect size as 0.25, the significance level as two-sided and 5%, and the power as 80%. We considered participants to be eligible if they were the family members of ischemic stroke or hemorrhagic stroke patients and had lived with and/or been the patient’s caregiver(s) since hospital admission of the patient for stroke. The participants had to be unpaid and provide assistance to support their loved one due to their underlying disability. Family caregivers could be immediate family caregivers/IFM (e.g., spouse, child, or sibling) or other relatives (in-laws, nephew/niece, cousin, or grandchild). Family caregivers were included if they could understand the Indonesian language and were willing to participate in this study. Nurses who performed a CDS assessment of stroke patients were nurses from the participating stroke ward who had a minimum of three years of experience working with stroke patients and had attended the CDS training session led by the first author (NN).

### 2.3. Data Collection

Data was collected from March 2016 to May 2016. The data collection was performed on the patient’s first day of care on the stroke care wards. The assessment of the degree of care dependency of the stroke patient on the first day of admission highlighted the effect of the stroke attack and a critical factor in the appropriate assignment of the nursing problem and nursing diagnosis. In the participating hospitals, the initial nursing admission assessment included gathering information concerning the patient’s care dependency with various forms. The family caregivers also had to prepare to enter the caregiving relationship by going through a training session about role engaging that often occurred right after hospital admission. Each participating family caregiver was instructed on how to fill in the CDS by trained nurses in the wards with a session lasting approximately fifteen minutes. The explanation was given in person and directly, either during an individual consultation session or during a group session with two to five family caregivers. Family caregivers received the CDS and an additional CDS instruction sheet. On the patient’s first day of care in the stroke care wards, the family caregivers were asked to fill out the CDS sheet and rate their relative’s care dependency level based on their perception of the current condition of the stroke care. Both nurses and family caregivers were requested to provide information on their age and gender on the demographic information sheet. In addition, family caregivers were asked to define their relationship to the respective stroke patient (e.g., spouse, sister, or cousin).

### 2.4. Instrument

The CDS contains fifteen items and is used to assess a patient’s level of care dependency in the areas of eating and drinking, continence, body posture, mobility, day and night patterns, getting dressed/undressed, body temperature, hygiene, avoiding danger, communication, contact with others, senses of rules, daily activities, recreational activities, and ability to learn [18]. In this study, the level of care dependency was assessed using the Indonesian version of the CDS. The Indonesian version of the CDS has an interrater reliability of 0.88 and an exact interrater agreement of 45.0%. The CDS assessments were scored using a 5-point Likert scale that ranged from 1 (completely care dependent) to 5 (almost care independent). Between 15 and 75 points per patient could be accrued. A lower CDS score indicated that the patient was more care dependent, whereas a higher CDS score indicated that the patient was more independent.

### 2.5. Data Analysis

Data that were related to the family caregivers’ assessments of the stroke patients’ levels of care dependency were compared with data from the nurses’ assessments of the respective stroke patients’ levels of care dependency. Data were analyzed using IBM SPSS 25 (Chicago, IL, USA). The participants’ characteristics were summarized in means for numerical data and percentages for categorical data. The data were tested for their normal distribution by applying the Kolmogorov–Smirnov test. The significance level was set to 5% to highlight suggestive evidence of differences between the caregivers’ assessments. A nonparametric analysis was performed using the Wilcoxon signed-rank test to determine the mean differences between the family caregivers’ and nurses’ assessments on the basis of the CDS sum score and items. The intraclass correlation coefficient (ICC) was tested as well.

### 2.6. Ethical Consideration

Ethical approval was granted to conduct this study by the responsible ethical committees. The institutional review board numbers for this study were 27-440 ex 14/15 from the Medical University of Graz, Austria, and 565/UN6.C1.32/Kep/PN/2015 from the Faculty of Medicine, Padjadjaran University, Indonesia. The participants received detailed information and gave informed consent manually.

## 3. Results

The demographic characteristics of the family caregivers and nurses are presented in Table 1. One hundred and twenty-five family caregivers were approached in this study, seven of whom refused to participate for personal or health reasons. Data from 118 family caregivers and 21 nurses were ultimately included in the study. Each family caregiver assessed the care dependency level of his or her loved one using the CDS. Participating nurses assessed four to six stroke patients each. An excellent intraclass correlation coefficient (0.97) was found between the care dependency scores of the family caregivers and nurses. The interrater reliability on 14 items ranged from 0.70 to 0.99, with only the item of eating and drinking revealing 0.15. Family caregivers and nurses were mostly female, and more than half of the family caregivers were the spouses of the stroke patients.

The outcome of the Wilcoxon signed-rank test revealed that significant differences existed between the family caregivers’ and the nurses’ care dependency sum scores for the stroke patients (*p* < 0.001): 38.3 (SD 15.7) and 44.3 (SD 20.0), respectively.

Table 2 shows that the majority of stroke patients were male, younger than 65 years old, and had suffered an ischemic stroke. The majority of the stroke patients had a comorbidity.

Table 3 shows the mean scores of the CDS items, which diverged on the family caregivers’ and nurses’ assessments. Nurses assigned higher scores on all CDS items than family caregivers, for example, for the item of eating and drinking, where nurses assigned an average score of 2.86 and family caregivers gave an average score of 2.21. Significant differences were observed between the family caregivers’ and nurses’ assessments on numerous items of the Care Dependency Scale. The highest r (effect size) was –0.49, which represented a medium to large difference (Cohen benchmark) between the nurses’ and family caregivers’ assessments. Only the communication item rates did not show any differences between the family caregivers’ and nurses’ assessments (z = −1.79, *p* > 0.05, r = −0.07).

## 4. Discussion

This was the first comparison of care dependency level between two different groups of caregivers for stroke survivors. Most family caregivers were identified as spouses of stroke patients. Several studies have reported that the spouse or partner frequently acts as the informal caregiver for a stroke patient, even in studies conducted in Western countries [19]. Other studies that were conducted in Asian countries (e.g., Thailand) reported that children of stroke patients most often acted as caregivers, even though the majority of the stroke patients were married [20]. This finding is supported by the fact that the cultural and traditional backgrounds of caregivers from these countries demand that children, and especially daughters or daughters-in-law, are to be responsible for their parents’ care [21]. In our study, children of stroke patients constituted the second highest group among family caregivers.

Our study revealed that the family caregivers’ and nurses’ assessment scores of stroke patients’ care dependency levels differ significantly. Nurses assessed the patients’ levels of care dependency lower than family caregivers did, which means that the nurses expected the post-stroke patient to do more than the family caregiver expected. This finding is in line with that of another study in which professional caregivers reported a lower prevalence of pain among nursing home residents than relatives did [22]. These pieces of evidence indicate that each type of rater has their own perspective. These different perspectives need to be recognized by health care practitioners [23], and in clinical situations, it is recommended for nurses to reach an agreement with family caregivers. However, our study revealed a high inter-rater reliability coefficient. This indicated that the ability of family caregivers and nurses to judge the care dependency level of stroke patients is stable and that the resulting scores are reliable.

Based on the results, we assumed that family caregivers may unintentionally trend toward offering a level of assistance that exceeds the patient’s needs. This may happen because of the family caregiver’s psychological status and relationship with the care recipient. A similar study that compared the unmet rehabilitation needs of stroke care patients and their caregivers’ perceptions of these needs showed that family caregivers perceived more unmet needs than the patients themselves did [24]. In another case, cancer patients in Indonesia reported that their family members supported their daily activities more frequently than nurses did [25]. It is suggested that nurses and family caregivers need to compare and discuss their perceptions of the care dependency levels of stroke patients. Nurses need to teach family caregivers to avoid providing too much assistance to their loved ones in order to help the stroke patient regain their independence. Furthermore, the family caregiver can use this approach to effectively communicate and come to an understanding with their care recipient about the proper level of assistance. This tactic will foster a sense of mutual understanding and sustain a manageable intervention level, which is an important aspect of providing stroke care at home [26].

Our findings on the differences between the family caregivers’ and nurses’ perceptions of the levels of care dependency may reflect that family caregivers perceive their relative as extremely ill, weak, and in need of their help. In fact, more than half of stroke patients in this study suffered from a comorbidity such as cardiovascular problems or diabetes, and the majority required high amounts of care. Another consideration is that the perceptions of family caregivers from Asian countries might be influenced by their cultural and religious values. In these countries, younger people are expected to be polite and offer assistance to their sick parents [3]. This phenomenon can also be observed in the relationship between wives and husbands. According to traditional Indonesian and Islamic values (Islam is the major religion in Indonesia), helping other people, especially one’s parents and spouse, is an obligation and is a highly valued deed. According to a study that explored the influence of culture on the family caregivers’ experiences, caregiving is a mandate of the Asian culture. Cultural norms strongly influence the decision to care for older persons in a family. Another reason why differences between the nurses’ and family caregivers’ assessments were noted is because nurses are professionally trained to assess patients and are skilled in identifying the patients’ needs. However, nurses have been advised to avoid making stereotypical judgments when conducting assessments.

This study revealed the effect size of a medium to large difference between CDS item scores rated by nurses and family caregivers. However, the findings of this study show that the participating caregivers and the nurses rated the care dependency item “communication” similarly. This might be indicative of the fact that the ability of a stroke patient to communicate with others is a clear feature of social contact and, therefore, easy for family caregivers to rate. In fact, at least thirteen care dependency items for stroke care recipients were assigned lower scores; both rater groups assessed higher care dependency for these items than for the “communication” item. This finding supports the assumption that the perception of the severity of care dependency constitutes an obstacle for family caregivers, influencing their assessment of other care dependency items. Thus, this study emphasized that nurses must give additional attention to a family caregiver who tends to help their care recipient beyond the care recipient’s need. Moreover, nurses must be ready to work together with family caregivers to reach a shared decision for the level of assistance so that stroke patients receive adequate support.

Every stroke case is different, and the respective care needs are also different. One patient may have retained all cognitive abilities but be heavily impaired in terms of speech and communication, while another may suffer from paralysis and need assistance with mobility and daily activities. Family caregivers may also react in different ways to the new situation which entails major changes in their lives and roles [27]. Some may feel afraid or overwhelmed by their caregiving tasks, whereas others are less troubled by having to assume new roles [28]. Therefore, it is vitally important that nurses and family caregivers closely cooperate and share information regarding the perceived level of care dependency. Nurses need to work together with the family caregivers to optimize the condition of the stroke patient’s post-discharge care at home [29]. Conducting an assessment of patient needs during the acute care phase represents the first step toward improving the readiness of family caregivers to step into their new roles [30]. This will help achieve the main goal: that the caregiver provides support only when and where it is truly needed, otherwise supporting the patient’s autonomy and allowing them to maintaining a sense of individual independence. Nurses need to provide information for family caregivers as early as possible to support discharge planning, in part due to the steadily increasing healthcare costs and reductions in the lengths of hospital stays, but also to improve the efficacy of the home care transition [6].

## 5. Conclusions

Although the assessments of the family caregivers and nurses were congruent, caregivers generally assessed the care dependency levels as higher than the nurses did. This piece of evidence may raise the nurses’ awareness of differences in perception, encouraging them to interact more with family caregivers and discuss the patient’s discharge planning as well as his or her post-discharge care. These study findings suggest the perceptions of both family caregivers and nurses regarding the care dependency needs of stroke care recipients should be incorporated into this planning. The Indonesian family caregivers the person in charge of post-discharge care needs to be encouraged to participate actively in the care dependency assessment. Since the data for this study were collected on the patients’ first days on a stroke ward, we highly recommend emphasizing and discussing the concept of and necessity for the care dependency assessment in more detail during the patient’s later care stages, for example, when their discharge from the hospital is imminent. By doing so, the quality of assistance can be improved in the home care setting in the future. More research on the utility of the CDS from nurses’ and family caregivers’ perspectives in other settings is still needed.

## Figures and Tables

**Table 1 healthcare-10-01007-t001:** Characteristics of family caregivers and nurses.

Characteristics	Family Caregiver M (SD) or *n* (%)	Nurse M (SD) or *n* (%)
Age	59.9 (12.25)	36.9 (9.16)
Sex		
Male	27 (22.9)	4 (19.1)
Female	91(77.1)	17 (80.9)
Family caregiver categories		
Spouse	67 (56.8)	
Child	41 (34.7)	
Sibling	6 (5.1)	
Other	4 (3.4)	

**Table 2 healthcare-10-01007-t002:** Characteristics of stroke patients.

Characteristics		*n*	%
Age	<65	76	64.4
	≥65	42	36.6
Sex	Male	77	65.3
	Female	41	34.7
Stroke type	Ischemic	85	72.0
	Hemorrhagic	33	28.0
Comorbidity	No	58	49.2
	Yes	60	50.8

**Table 3 healthcare-10-01007-t003:** Differences among care dependency items between raters.

CDS Items	Family Caregivers	Nurses	z	Significance (*p*)	r
	Mean (SD)	Mean (SD)			
Eating and drinking	2.21 (1.20)	2.86 (1.57)	−6.62	0.000	−0.43
Continence	2.30 (1.20)	3.12 (1.50)	−7.59	0.000	−0.49
Body posture	2.36 (1.06)	3.00 (1.34)	−6.33	0.000	−0.41
Mobility	2.32 (1.19)	2.66 (1.46)	−5.41	0.000	−0.35
Day and night pattern	3.36 (1.60)	3.40 (1.61)	−2.24	0.025	−0.14
Getting dressed	2.10 (1.08)	2.55 (1.44)	−6.31	0.000	−0.41
Body temperature	2.64 (1.25)	3.26 (1.56)	−7.43	0.000	−0.48
Hygiene	2.14 (1.08)	2.48 (1.39)	−5.22	0.000	−0.34
Avoiding danger	2.41 (1.20)	2.91 (1.53)	−6.33	0.000	−0.41
Communication	3.35 (1.56)	3.43 (1.62)	−1.79	0.073	−0.07
Contact with others	3.14 (1.46)	3.30 (1.57)	−4.24	0.000	−0.27
Sense of rules	3.19 (1.50)	3.30 (1.58)	−3.61	0.000	−0.23
Daily activities	2.45 (1.31)	2.68 (1.45)	−4.24	0.000	−0.27
Recreation activities	2.29 (1.28)	2.55 (1.48)	−4.80	0.000	−0.31
Ability to learn	2.04 (1.08)	2.81 (1.53)	−6.41	0.000	−0.42

## Data Availability

The research data associated with this paper is available through the permission from the authors.

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
