# Peer review of "Care Dependency of Hospitalized Stroke Patients Based on Family Caregivers’ and Nurses’ Assessments: A Comparative Study"

_healthcare, 2022, doi:10.3390/healthcare10061007_

Round 1

Reviewer 1 Report

Initially, I congratulate the authors for choosing the topic, current and relevant.

It is important to empower family caregivers to properly care for the dependent family person.

 The authors present and discuss the object of study, describe the methodology used, and the results are clearly described and discussed in the light of current and relevant references.

 I recommend a more detailed description of the method, for example, the total duration of the session, who made the session, etc

Materials and methods

Line 136 – more space

Line 139 -140- Clarify who made the session and for how long

 Line 142- clarify when the family caregivers fill out the CDS sheet

Line 145 the demographic information sheet only had age, gender, and relation?

Author Response

Thank you for the valuable suggestion.

Reviewer 2 Report

Thank  you for this interesting study of the difference in care dependency ratings between nurses and family caregivers in Indonesia using the CDS. 

There were a couple of spots I would suggest edits.  Line 52 looks like an incomplete thought.  What do you mean by insufficient social support?  It's an important thought, so could you add a little more? 

Line 192 Table 1 has a problem.  It looks like Table 2 was accidently copied into Table 1?  There is no heading.

Line 200 has a typo, easy fix.  "caregivers"". 

In the results, you point to Table 1 for demographics. I see that you mention comorbidity, but you don't tell the reader what comorbidity is.  Perhaps you might like to explain comorbidity, because you bring it up again in the discussion in line 250. 

I appreciate that stroke patients and caregivers respond differently to care needs in the discussion, and the cultural  norms addressed.

Thank you

Author Response

Thank you for the valuable suggestion.

This manuscript is a resubmission of an earlier submission. The following is a list of the peer review reports and author responses from that submission.